# The Effects of the Dietary Approaches to Stop Hypertension (DASH) Diet on Metabolic Syndrome in Hospitalized Schizophrenic Patients: A Randomized Controlled Trial

**DOI:** 10.3390/nu11122950

**Published:** 2019-12-04

**Authors:** Tamara Sorić, Mladen Mavar, Ivana Rumbak

**Affiliations:** 1Psychiatric Hospital Ugljan, Otočkih dragovoljaca 42, 23275 Ugljan, Croatia; ravnatelj@pbu.hr; 2Faculty of Food Technology and Biotechnology, University of Zagreb, 10000 Zagreb, Croatia; ivana.rumbak@pbf.hr

**Keywords:** DASH diet, diet quality, metabolic syndrome, nutrition counseling, schizophrenia

## Abstract

The relationship between the Dietary Approaches to Stop Hypertension (DASH) diet and metabolic syndrome (MetS) in people with schizophrenia is unknown and remains to be investigated. Therefore, we have conducted a three-month parallel-group randomized controlled trial. Sixty-seven hospitalized schizophrenic patients with MetS [*n* = 33 in the intervention group (IG) and *n* = 34 in the control group (CG)] completed the intervention. The IG followed the DASH diet with the caloric restriction of approximately 1673.6 kJ/day (400 kcal/day) when compared to the standard hospital diet followed by the CG. Simultaneously, both groups participated in a nutrition counseling program. Anthropometric and biochemical parameters and blood pressure were measured at the baseline and after three months, while nutrient intakes during the intervention were assessed using three non-consecutive 24-hour dietary recalls. The analyses were carried out based on the per-protocol approach. At three months, the MetS prevalence significantly decreased in both the IG and the CG (75.8%, *p* = 0.002, and 67.7%, *p* = 0.0003, respectively; odds ratio = 0.9; 95% confidence interval = 0.43–1.87). No significant differences in the prevalence of MetS and its features were found between the groups.

## 1. Introduction

The metabolic syndrome (MetS) is a complex medical condition consisting of several interconnected parameters including abdominal obesity, elevated serum triglycerides (TG), decreased levels of high-density lipoprotein cholesterol (HDL-C), elevated blood pressure (BP), and increased levels of fasting blood glucose (GLC) [1], and is considered as one of the major predictors for the development of cardiovascular diseases and type 2 diabetes mellitus [1,2]. Due to continuous increase in MetS prevalence during the last few decades [3], it has become one of the leading global public health problems [4]. However, in certain population groups, including people with serious mental illnesses such as schizophrenia, MetS prevalence is estimated to be much higher than among people without mental disorders [5]. According to the results of numerous previously conducted studies, the MetS prevalence in people with schizophrenia is approximately 1.5–2 times higher than among the mentally healthy people [6,7]. Furthermore, it is already well-known that people suffering from schizophrenia have reduced life expectancy and increased mortality rate when compared to the general population [8], which is largely due to the presence of MetS [9].

Because of the aforementioned, the syndrome itself and the identification of possible mechanisms for its prevention and treatment have been in the focus of numerous researches in the field of psychiatry. Despite the scientific interest, the exact reasons for a higher occurrence of MetS among people with schizophrenia have not yet been completely clarified [9]. Although atypical antipsychotics are considered to be the major contributor [10], there is a growing body of evidence suggesting that unhealthy dietary habits, common for individuals suffering from schizophrenia, have a significant role in its development [9,11]. Previously published studies conducted among people with schizophrenia reported higher intake of salt, total sugars, and fats, primarily saturated fatty acids (SFA), along with the lower intake of dietary fiber and fruits when compared to the general population [12,13,14,15]. Even though such dietary habits are generally associated with unfavorable health implications [16,17,18,19], there is still no consensus on the most effective nutrition therapy for the MetS [20]. Therefore, it is of an utmost importance to develop an adequate nutritional intervention program that would be effective in its treatment.

The Dietary Approaches to Stop Hypertension (DASH) is a diet primarily designed to lower BP [21,22]. In a DASH dietary pattern, a special emphasis has been put on low-fat milk and dairy products, whole grains, lean meat, fruits, and vegetables, together with the reduction of sodium intake [23]. Although primarily intended for individuals with the diagnosis of hypertension, according to Hill et al., it could also be beneficial in treating MetS [24]. Until today, several randomized controlled trials (RCTs) examined the impact of the DASH diet on MetS or some of its features in different population groups, and the results mostly showed amelioration of the studied parameters [25,26,27]. Beneficial effects of the DASH dietary pattern on cardiometabolic parameters in people with and without diabetes mellitus were also confirmed in the recently conducted umbrella review by Chiavaroli et al. [28]. However, studies on the relationship between the DASH diet and different health outcomes in people with schizophrenia are still insufficient and limited [29] and, to the best of our knowledge, there are no published RCTs evaluating the effects of the DASH diet on MetS in this specific population group.

Therefore, the main purpose of the present study was to examine the effects of the DASH diet on MetS and its features in hospitalized schizophrenic patients. In addition, we also examined the effects of this specific dietary pattern on other studied anthropometric and biochemical parameters and on the diet quality. We hypothesized that the application of the DASH diet would result in significant improvements in all of the studied health parameters, as well as in the diet quality.

## 2. Materials and Methods 

### 2.1. Study Participants and Setting

The present three-month parallel-group RCT was conducted between May 2, 2017 and December 20, 2017 in the Psychiatric Hospital Ugljan, Croatia. 

Hospitalized individuals, both men and women, aged between 18–67, with the diagnosis of schizophrenia (F20.0–F20.9) according to the 10th Revision of the International Classification of Diseases and without any physical or cognitive impairments that could disable full participation in the study were screened after providing written informed consent to verify whether they meet all of the other inclusion criteria for the participation in the RCT. All the study participants had previously been hospitalized and were residing in the Psychiatric Hospital Ugljan prior to commencement of the study because of the exacerbation in their mental illness, and were considered eligible to enter the study only if they were in a stable phase of schizophrenia (defined by the fact that the prescribed antipsychotic therapy has not changed significantly in the previous month). Other inclusion criteria were as follows: having a diagnosis of MetS according to the Joint Interim Statement definition [30], taking antipsychotic medications for the last six months or longer, and providing a written informed consent for the participation in the RCT. On the other hand, individuals were considered ineligible to participate in the RCT if they followed one of the specific hospital diets with restrictions in the intake of specific food items, groups and/or nutrients, if they were currently taking weight loss pharmacotherapy, or if they experienced a significant body weight (BW) loss in the last three months. Participants were also excluded from the study for one or more of the following reasons: personal request, exacerbation in mental condition, the appearance of a new illness that could disable full participation or could have an impact on the study outcomes, a significant change in pharmacotherapy, cognitive deteriorations that disable full participation, disinterest and refusal to fully participate, or hospital discharge. The researcher (hospital nutritionist) recruited all the study participants by word-of-mouth. 

Sample size calculation was performed using a module Power analysis Sample Size Calculation of the statistical software Statistica, version 6.1. (StatSoft Inc, Tulsa, OK, USA) [31]. The calculation was based on the impact of nutritional intervention on MetS and its components, for the two intervention arms. The MetS prevalence at three months was the primary endpoint for the sample size calculation and the calculation was based on the expectation that the three-month intervention program would result in the reduction of MetS prevalence from 100% to 57% in the intervention group (IG) and in the reduction of MetS prevalence from 100% to 90% in the control group (CG). Sample size calculation was also based on the fact that between-group and within-group difference of 10% in the following parameters is considered clinically significant: BP, TG, HDL-C, low-density lipoprotein cholesterol (LDL-C), total cholesterol (TC), and GLC. With the estimated magnitude of significant difference of 10%, a standard deviation of 15%, and with a type I error rate (alpha) 0.05 and a power goal of 0.80, the threshold was 33 participants per group. The obtained value was increased by 15%–20% due to the assumed drop-outs.

### 2.2. Randomization

The participants who met all the inclusion criteria were randomized to the IG or the CG using computer-generated random numbers (Excel RAND function). In a stratified randomization scheme, the study participants were block-randomized by age, sex, body mass index (BMI), and waist circumference (WC) at a ratio of 1:1. The randomization process was performed by an independent statistician and the randomization list was then provided to the researcher who assigned participants to the study groups. Both the researcher and the participants were not blinded to the study allocation. 

### 2.3. Ethics

Participation in the study was entirely voluntary and all participants had to provide written informed consents, separately for participation in the pre-enrollment screening and in the RCT. Out of 79 randomized participants, 22 participants (*n* = 11 in the IG and *n* = 11 in the CG) were deprived of legal capacity. For those participants, written informed consents were provided from both the participants and their legal guardians. The number of participants deprived of legal capacity who have completed the three-month intervention is presented in Table 1. Prior to signing of the informed consent, the researcher thoroughly explained the study procedures, aims, and potential outcomes individually to each participant and his legal guardian, where applicable. The study was approved by the Ethics Committee of the Psychiatric Hospital Ugljan (approval number: 01-552/01-16; approval date: November 21, 2016) and the Central Ethics Committee of the Medical School, University of Zagreb (approval number: 380-59-10106-17-100/56; approval date: February 23, 2017). All the study procedures were performed following the principles of the Declaration of Helsinki. The clinical trial was registered in ClinicalTrials.gov (NCT04025073).

### 2.4. Study Design

During the three-month intervention period, the IG was requested to follow the DASH diet. The caloric intake of the prescribed diet was reduced by approximately 1673.6 kJ/day (400 kcal/day) when compared to the standard hospital diet (SHD). Higher caloric restriction, even though it could lead to better outcomes, was circumvented in order to prevent its potential negative psychological side effects. The diet was planned by the researcher according to the specific DASH diet recommendations and prescribed macronutrient and micronutrient contents [23]. The average daily intakes of the most important components of the prescribed DASH diet were: energy intake 8067.6 kJ (1928.2 kcal), total fat 24.5%, SFA 5.8%, protein 19.4%, carbohydrates 55.7%, cholesterol 111.8 mg, sodium 2170.1 mg, potassium 4769.3 mg, calcium 1262.6 mg, magnesium 501.1 mg, and dietary fiber 31.0 g. Simultaneously, the CG continued to follow the SHD planned according to the requirements determined by the decision on the nutritional standard for hospitalized patients in Croatia [32]. The calculated average daily intakes of the same nutrition components indicated for the DASH diet were as follows: energy intake 9770.1 kJ (2335.1 kcal), total fat 28.3%, SFA 7.8%, protein 16.8%, carbohydrates 54.8%, cholesterol 222.3 mg, sodium 3905.5 mg, potassium 3866.7 mg, calcium 1284.23 mg, magnesium 359.4 mg, and dietary fiber 24.6 g. Besides the aforementioned, in the DASH diet the refined cereals (white bread, white rice, and white pasta) were replaced with whole grains (whole-grain bread, brown rice, and whole-wheat pasta), while low-fat dairy products (0.9% fat content) replaced those with 2.8% fat content. Nuts and seeds, together with fruits and vegetables occupied an important place in the DASH daily menus. For meal planning and nutrient composition analyses computer software Program Prehrane 5.0 (IG PROG, Rijeka, Croatia) has been used. Meal planning and nutrient composition analyses were conducted by the researcher.

Additionally, both the IG and the CG participants joined the nutrition counseling program designed to give them all relevant information about healthy eating. The program included the following four lectures: MyPlate dietary guidelines [33]; The basic principles of healthy, balanced nutrition; How to read and understand food labels?, and Diet and chronic diseases―prevention and treatment. Because of the interactive character, the lectures were organized in groups of up to 10 participants. At the end of the program, participants were given the brochures summarizing the most important information covered by the lectures. 

### 2.5. Data Collection and Outcome Measures

Participants’ basic socio-demographic and lifestyle characteristics were gathered by the researcher in a form of face-to-face interviews, while the relevant medical history details were provided by the hospital personnel. The global severity of the psychotic illness was evaluated at the baseline by hospital psychiatrists using the Signs and Symptoms of Psychotic Illness (SSPI) scale [34]. In order to estimate an additional daily energy intake in the previous month, the participants were asked to report the consumption frequency, along with an estimated quantity and name of the manufacturer, of all the foods and beverages individually purchased and consumed in addition to the ones regularly provided by the hospital. The interviews lasted approximately 15 minutes and were carried out by the researcher at the beginning of the intervention. All the gathered information was subsequently checked with the medical personnel of the departments where the participants were located. The calculations of additional daily energy intakes were done using the computer software Program Prehrane 5.0.

During the intervention, compliance with the prescribed diets (both the DASH diet and the SHD) was assessed using three non-consecutive 24-hour dietary recalls (24-h DRs) (two weekdays and one weekend day) for each participant. DRs were administered by the researcher, and the participants were asked to recall all the foods and beverages, regardless of whether they were provided by the hospital or individually purchased, together with their estimated quantities, consumed in the previous day. Standard household measures such as cups, spoons, and serving sizes were used to help participants estimate the amounts of foods and beverages consumed. The average duration of the single interview was approximately 20 minutes. On average, three 24-h DRs were conducted daily in order to cover as much intervention days as possible. For each participant, one 24-h DR was assessed per month. Due to the additional consumption of the purchased foods and beverages, which could not be restricted, when conducting the 24-h DRs, it was of the utmost importance that the participants were adherent to either the DASH diet or the SHD. Those participants refusing the prescribed diets provided by the hospital would have been excluded from the study. The analyses of the 24-h DRs were performed by the researcher using the computer software Program Prehrane 5.0. As the intervention was conducted in an inpatient setting, knowing the meal recipes was material for precise quantification of the consumed food portions and for the calculation of nutritional intakes. For the items not present in the Program Prehrane 5.0 database, the food composition data provided on food labels were used. 

Furthermore, basic anthropometric measurements including measurements of BW, body height (BH), WC, hip circumference (HC), and % body fat (% BF) were carried out by the researcher following the standard instructions [35]. BW and BH were measured using a digital medical scale with stadiometer Seca 799 (Vogel & Halke GmbH & Co., Hamburg, Germany). BMI was calculated using the following formula: BW (kg) / BH (m^2^). WC and HC were measured using a non-stretchable measuring tape and the measured values were used for the calculation of the WHR by the formula: WC (cm) / HC (cm). The OMRON BF500 analyzer (Omron Healthcare, Vernon Hills, IL, USA) was used for the determination of % BF. 

Blood samples were taken after an overnight fast from the antecubital vein and all the relevant concentrations were determined on the day of the blood sampling. Blood GLC concentration was measured using the enzymatic reference method with hexokinase. Serum TG and TC levels were measured using the enzymatic colorimetric method, while the homogeneous enzymatic colorimetric test has been used for the determination of serum HDL-C concentrations. The analyses were performed using commercially available enzymatic reagents on the Cobas c 111 analyzer (Roche Diagnostics Ltd., Rotkreuz, Switzerland). LDL-C concentrations were calculated using the Fiedewald formula [36]. Blood samplings, together with blood sample analyses and calculations were carried out by laboratory personnel in the Department of Medical Biochemistry, Psychiatric Hospital Ugljan.

Two trained nurses from the Department of Psychosomatic Medicine, Psychiatric Hospital Ugljan performed two consecutive readings of BP with the five-minute interval between the readings using the aneroid BP gauge Riester Ri-san (Rudolf Riester GmbH, Jungingen, Germany). The readings were carried out following the standard instructions [37]. For the analysis, the mean value of the two readings has been used.

Anthropometric measurements, blood sampling, blood sample analysis, and BP assessment were conducted both at the baseline and at the end of the intervention.

According to the Joint Interim Statement definition [30], participants were determined to have MetS if they had at least three of the following components: (1) elevated WC (cut-off points for Europids: ≥94 cm for men, ≥80 cm for women); (2) elevated TG (≥1.7 mmol/L) or drug treatment for elevated TG; (3) reduced HDL-C (<1.0 mmol/L for men, <1.3 mmol/L for women) or drug treatment for reduced HDL-C; (4) elevated BP [systolic blood pressure (SBP) ≥130 mmHg and/or diastolic blood pressure (DBP) ≥85 mmHg] or antihypertensive drug treatment in a patient with a history of hypertension; (5) elevated fasting GLC (≥5.6 mmol/L) or drug treatment for elevated GLC.

### 2.6. Statistical Analysis

Statistical analyses were performed using statistical software Statistica, version 6.1 (StatSoft Inc, Tulsa, OK, USA). The assumption of normality was investigated with the Kolmogorov–Smirnov test and the Shapiro–Wilk W-test. Due to the drop-outs, the analyses were carried out based on the per-protocol approach.

Participants’ basic characteristics, medical history details, and nutrient intakes were presented as means and standard deviations (SD) (metric data) or as numbers of participants and percentages (nominal/ordinal data). Pre- and post-intervention numerical results were shown as means and SD, along with 95% confidence intervals (CIs), while pre- and post-intervention categorical data were presented as numbers of participants and percentages, along with 95% CIs for one proportion calculated using the Pearson Chi-square test. To understand the size of changes found, estimated effects and the respective 95% CIs were calculated. To evaluate the between-group differences in basic characteristics and medical history details, the paired samples t-test and Mann–Whitney U test were used for numerical variables, and the Pearson Chi-square test was used for categorical variables. Pre- and post-intervention numerical data were compared within the groups using a paired samples t-test and between the groups using an independent samples t-test. For the comparison of pre- and post-intervention categorical data within and between the groups, a Pearson Chi-square test or a Fischer’s exact test were used. To analyze the between-group differences in the nutrient intakes, a paired samples t-test was used. The threshold of significance for all the analyses was *p* < 0.05.

## 3. Results

Out of 79 participants allocated either to the IG or the CG, a total of 67 participants (33 participants in the IG and 34 participants in the CG) completed the three-month intervention. The reasons for participants’ drop-out were hospital discharge (*n* = 11) and death (*n* = 1) (Figure 1). The participation in the present study could not influence the length of hospital stay and the participants “ready to go home” were discharged regardless of the MetS and their participation in the RCT. Discharged participants were excluded from analysis because all the relevant data were not obtained prior to hospital discharge. During the intervention, there were no unintended effects reported. 

Participants’ basic characteristics and medical history details are summarized in Table 1. The vast majority of the participants were men [28 participants (84.8%) in the IG and 29 participants (85.3%) in the CG]. The mean age of the IG participants was 53.2 ± 8.9 years and of the CG participants 50.7 ± 8.0 years. There were no significant differences detected between the IG and the CG regarding any of the studied parameters listed in Table 1, except for taking hyperglycemia medications (*p* = 0.028). 

The impact of the DASH diet vs. the SHD on studied anthropometric and biochemical parameters and BP are summarized in Table 2. Compliance with the DASH diet resulted in significant decrease in BW (mean difference (MD) ± SD = 2.43 ± 3.14; 95% CI = 1.36–3.50; *p* < 0.001), BMI (MD ± SD = 0.79 ± 1.08; 95% CI = 0.43–1.16; *p* < 0.001), WC (MD ± SD = 3.54 ± 3.84; 95% CI = 2.23–4.85; *p* < 0.001), HC (MD ± SD = 1.76 ± 2.44; 95% CI = 0.92–2.59; *p* < 0.001), WHR (MD ± SD = 0.02 ± 0.04; 95% CI = 0.004–0.03; *p* = 0.015), and DBP (MD ± SD = 5.30 ± 9.78; 95% CI = 1.97–8.64; *p* = 0.004) in the IG. On the other hand, following the SHD resulted in significant decrease in the values of the following parameters among the CG participants: BW (MD ± SD = 1.65 ± 3.85; 95% CI = 0.35–2.94; *p* = 0.018), BMI (MD ± SD = 0.50 ± 1.19; 95% CI = 0.10–0.90; *p* = 0.019), WC (MD ± SD = 2.45 ± 4.18; 95% CI = 1.05–3.86; *p* = 0.002), and WHR (MD ± SD = 0.01 ± 0.03; 95% CI = 0.01–0.02; *p* = 0.006). When comparing the two groups, there were no significant differences detected for any of the parameters listed in Table 2, both at the baseline and at the end of the intervention period.

The impact of the DASH diet on the prevalence of MetS and its features are shown in Table 3. At three months, a significant decrease in MetS prevalence was observed in both the IG and the CG (75.8%, *p* = 0.002 and 67.7%, *p* = 0.0003, respectively; OR = 0.9; 95% CI = 0.43–1.87). However, we did not detect any significant between-group differences in the prevalence of MetS and its features after the intervention.

Finally, we have examined the impact of the intervention on diet quality. Compliance with the DASH diet and the SHD was assessed using a total of 201 24-h DRs (99 for the compliance with the DASH diet and 102 for the compliance with the SHD). For both the IG and the CG, the average daily intakes of the most important nutrient components in the DASH dietary pattern are presented in Figure 2. During the intervention, total daily intakes of energy, cholesterol, and sodium were significantly lower in the IG, when compared to the CG (all *p* < 0.001). Furthermore, the participants in the IG consumed significantly more dietary fiber, potassium, and magnesium (all *p* < 0.001) than in the CG. Regarding the percentages of energy from macronutrients, the diet of the IG participants had a significantly lower percentage of energy derived from total fats and SFA (all *p* < 0.001), along with a significantly higher percentage of energy from carbohydrates (*p* < 0.001) and protein (*p* = 0.001). For the total daily intake of calcium, no significant difference was noted between the two studied groups (*p* = 0.946).

## 4. Discussion

The present research, which was based on the per-protocol approach, showed no significant differences in the prevalence of MetS and its features between the IG and the CG. In addition, we have found that the application of the DASH diet, when compared to the SHD, significantly improved the diet quality of hospitalized schizophrenic patients with MetS. To the best of our knowledge, this is the first RCT which aimed to evaluate the impact of the DASH diet on MetS and its features in hospitalized schizophrenic patients.

Due to its high prevalence and numerous proven negative health implications, MetS represents one of the major concerns among people suffering from schizophrenia. Although the occurrence of MetS in people with schizophrenia has been mainly associated with antipsychotic treatment and characteristics of the psychotic illness, several general factors, including inadequate dietary habits, are also known for enhancing MetS predispositions [38]. Thus, asserting healthy dietary habits needs to be considered as one of the key factors in the treatment of MetS among people with schizophrenia. However, the RCTs examining the effects of nutritional interventions on MetS in people with schizophrenia are lacking and inconsistent, and it is, therefore, still unclear what would be the most appropriate nutritional approach for its treatment. In the last few decades, the DASH dietary pattern has been in the focus of numerous researches which have confirmed its favorable effects not only on BP, but also on MetS features in different population groups [25,26,39]. Nevertheless, no RCTs on the association between the DASH diet and MetS and its features in hospitalized schizophrenic patients have been published so far. In the present study we have detected significant improvements in several anthropometric parameters, including BW, BMI, WC, and WHR, in both the IG and the CG. These results are of great importance because the improvement in all of the aforementioned anthropometric parameters is crucial for the effective treatment of MetS [40]. 

After three months, significant decrease in DBP was observed among the participants in the IG, when compared to the baseline values. On the other hand, we failed to detect significant improvements in the SBP. When compared to the CG, there were no significant differences in the DBP and SBP levels at the end of the intervention. Our results are, therefore, not fully consistent with the findings of Saneei et al. [25]. In their RCT conducted on adolescents, Saneei et al. revealed that following the DASH diet recommendations, when compared to the general dietary advice, prevented the increase in DBP, while at the same time not affecting SBP levels [25]. Contrary to what we hypothesized at the beginning, we have failed to find significant improvements in the serum glucose levels and blood lipid profile in the IG after completing the three-month intervention. Thus, the aforementioned results are inconsistent with those of previously conducted studies [26,41,42]. 

The IG experienced a significant reduction in the MetS prevalence after three months. Surprisingly, what it was not expected is that the present study would result in the significant reduction of the MetS prevalence among the CG participants as well. As numerous confounding factors (Table 1) were taken in consideration, the improvements observed in the CG could potentially be due to the impact of some nonspecific factors, including the awareness of the study participants about the special emphasis being put on their diet, higher intention provided to participants during the intervention, and the desire to please the researcher [43]. Unfortunately, the present study concept did not allow us to conclude whether the impact of the aforementioned nonspecific factors influenced the positive outcomes observed in the CG. 

In the CG, the prevalence of elevated BP significantly increased by the end of the intervention. Even though the between-group difference in the prevalence of elevated BP was not statistically significant, the results suggest that the DASH diet could potentially slow down the development of hypertension. Beneficial effects of the DASH diet on the prevalence of MetS and its features have been confirmed in several studies conducted on other population groups. Following the DASH diet recommendations in the population of adolescent girls resulted with significant reductions in the prevalence of elevated BP and MetS [25]. Specifically speaking, in the terms of the MetS prevalence, following the DASH dietary advice for six weeks reduced the prevalence from 56.7% to 46.7%, while the prevalence of the participants following usual dietary guidelines has not changed (53.3% both at baseline and at 6th week). Furthermore, the within-group change in the prevalence of the high BP did not reach statistical significance among the adolescent girls following the DASH diet (35% at baseline vs. 30% at 6th week), but when comparing the prevalence from the DASH diet group and the group following usual dietary guidelines the difference was statistically significant after six weeks (30% vs. 43%, respectively) [25]. In another RCT conducted on adult outpatients with the diagnosis of MetS, the adherence to the individually prescribed DASH diet led to the significant decrease in MetS prevalence (from 100% at baseline to 65% at six months), along with the significant ameliorations in MetS features when compared to the control diet and standard weight-reducing diet. The group following the weigh-reducing diet decreased the prevalence of MetS from 100% at baseline to 81% at six months, while the group following the control diet experienced no changes in the MetS prevalence [26]. Contrary to the findings of the aforementioned studies, the effectiveness of the DASH diet, when compared to the SHD, was not proven in the present study.

The results of the 24-h DRs have shown certain deviations from the prescribed nutrient intakes in both the IG and the CG. The observed discrepancies were mainly caused by the consumption of individually purchased food items. During the RCT, those purchasing habits of both groups had not changed significantly in the terms of the additional daily energy intake observed before the beginning of the intervention (the data not shown in the present study). Despite the observed deviations, the average daily energy intake of the IG remained reduced by approximately 1601.2 kJ (382.7 kcal) when compared to the average daily energy intake of the CG. Along with the energy intake, we have found that the application of the DASH diet, when compared to the SHD, led to significant reductions in the total daily intake of cholesterol, sodium, and percentages of energy from total fats and SFA, while it significantly increased the intake of dietary fiber, potassium, and magnesium. Despite the imperfect compliance, nutrient intakes of the IG and the CG remained significantly different in all of the main features that constitute the DASH diet, except for the intake of calcium (both the IG and the CG have reached the DASH goal for calcium intake). Therefore, the aforementioned findings suggest that this type of nutritional intervention is a beneficial tool for diet quality improvement in people with schizophrenia. Similar to our findings, in the research conducted by Teasdale et al. [44], a 12-week dietary intervention led to significant reductions in the daily intakes of energy and sodium in young people with first-episode psychosis, while Baker et al.’s [45] telephone intervention program significantly improved the overall diet quality in people with psychotic disorders. 

The very nature of the participants’ mental illness is of relevance for the interpretation of the obtained results. People with schizophrenia are generally less motivated to change unhealthy lifestyle patterns, including poor dietary habits. Hence, this could represent one of the possible reasons for discrepancies found between the prescribed and actual nutrient intakes which could consequently affect the main outcomes of the present study. 

The key strength of our study is that it was conducted in an inpatient psychiatric setting where all the participants were provided with fully prepared meals throughout the intervention, and which has, in a great extent, reduced a participation burden. There were no significant differences between the IG and the CG considering a wide range of parameters (Table 1) that could potentially have an impact on the study outcomes. Therefore, the confounding effects of the selected parameters on the study outcomes are unlikely. Additionally, the design of the intervention program enables its reproduction in a large variety of hospital and nursing home facilities which would allow for results to be verified. 

There are few study limitations that ought to be considered. Although some of the previously conducted studies [25,42] resulted in significant improvements over even a shorter period of time, the key reason for the absence of significant differences between the studied groups at the end of the intervention period in this study might have been due to a limited time duration of three months. Furthermore, even though the RCT was conducted in a controlled hospital environment, we could not restrict participants’ individual purchasing of foods and beverages that were consumed in addition to the ones provided as a part of the intervention program. Additional energy intake was considered as one of the confounding factors, however we have not determined to what extent deviations caused by additional foods and beverages consumption influenced the final results. Finally, the present study did not account for some nonspecific factors mentioned earlier in the Discussion that could potentially explain the improvements observed in the CG and the absence of significant results between the groups.

Future studies with longer intervention duration are needed. Those studies should include additional study group that will receive neither the DASH diet nor the nutrition counseling in order to precisely assess their separated impacts on MetS and its features. For future studies it is also of essential importance to take in consideration all the nonspecific factors that could have the impact on the study’s findings. Additionally, in those studies, a personal nutritional approach should be considered, as the adjusting of energy and nutrient requirements at the individual level are likely to lead to better results. 

## 5. Conclusions

In conclusion, the DASH diet, when compared to the SHD, did not significantly affect MetS and its features, while it significantly improved the diet quality of hospitalized schizophrenic patients with MetS. 

## Figures and Tables

**Figure 1 nutrients-11-02950-f001:**
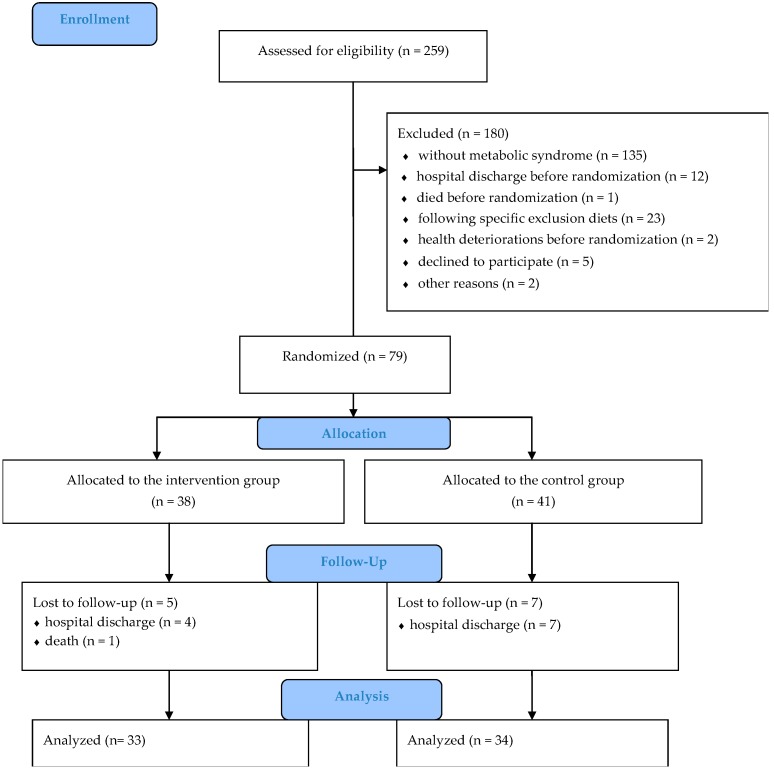
The CONSORT flow diagram showing the progression by stages of a randomized controlled trial.

**Figure 2 nutrients-11-02950-f002:**
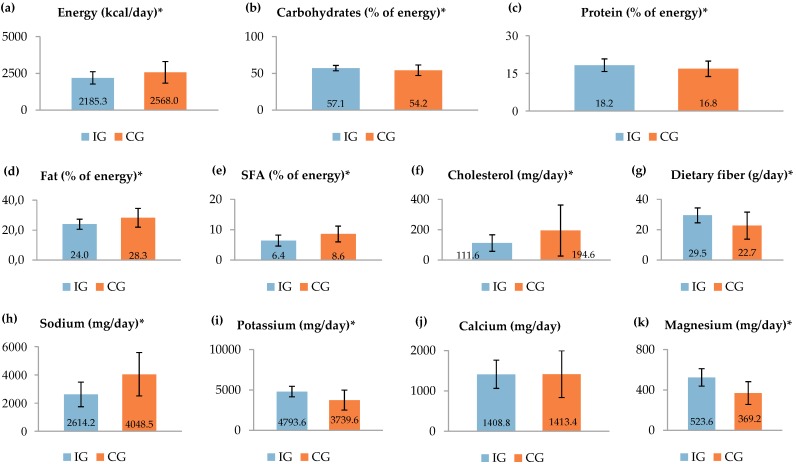
Nutrient intakes of the intervention group (IG = 33) and the control group (CG = 34) during the three-month intervention. Nutrient intakes are presented as means with standard deviations within bars. The first graph shows the energy intake in kcal/day. The values are equal to 9143.3 ± 1748.2 kJ/day and 10744.6 ± 3091.0 kJ/day, respectively (conversion factor: 1 kcal = 4.184 kJ). * significant difference (*p* < 0.05; calculated using a paired samples t-test): (**a**) *p* < 0.001; (**b**) *p* < 0.001; (**c**) *p* = 0.001; (**d**) *p* < 0.001; (**e**) *p* < 0.001, (**f**) *p* < 0.001; (**g**) *p* < 0.001, (**h**) *p* < 0.001; (**i**) *p* < 0.001; (**j**) *p* = 0.946; (**k**) *p* < 0.001.

**Table 1 nutrients-11-02950-t001:** Basic characteristics and medical history details of the study participants.

Characteristics	IG (*n* = 33) *n* (%) or Mean (SD)	CG (*n* = 34) *n* (%) or Mean (SD)	*p*
Sex			
male	28 (84.8)	29 (85.3)	0.959 ^a^
female	5 (15.2)	5 (14.7)	
Age (years)	53.2 (8.9)	50.7 (8.0)	0.228 ^b^
Educational level			
no formal education or primary education	12 (36.4)	7 (20.6)	0.568 ^a^
secondary education or higher	21 (63.6)	27 (79.4)	
Working status			
employed	0 (0) ^d^	1 (2.9) ^d^	0.450 ^a^
unemployed	15 (45.5)	11 (32.4)	
retired	17 (51.5)	19 (55.9)	
receiver of social welfare	1 (3.0)	3 (8.8)	
Marital status			
single	29 (87.9)	31 (91.2)	0.967 ^a^
married/in a relationship	4 (12.1)	3 (8.8)	
Residential area			
urban	18 (54.5)	11 (32.4)	0.113 ^a^
rural	15 (45.5)	23 (67.6)	
Illness duration (years)	17.6 (12.7)	17.9 (12.6)	0.808 ^c^
Type of schizophrenia			
paranoid	22 (66.7)	27 (79.4)	0.171 ^a^
residual	11 (33.3)	5 (14.7)	
other	0 (0) ^d^	2 (5.9) ^d^	
Deprivation of legal capacity			
yes	10 (30.3)	10 (29.4)	0.851 ^a^
no	23 (69.7)	24 (70.6)	
Number of antipsychotics			
1	3 (9.1)	9 (26.5)	0.199 ^c^
2–3	27 (81.8)	22 (64.7)	
>3	3 (9.1)	3 (8.8)	
Type of antipsychotics			
typical	1 (3.0)	2 (5.9)	0.511 ^a^
atypical	20 (60.6)	16 (47.1)	
both	12 (36.4)	16 (47.1)	
Taking antidepressants			
yes	3 (9.1)	4 (11.8)	0.721 ^a^
no	30 (90.9)	30 (88.2)	
Taking mood stabilizers			
yes	13 (39.4)	13 (38.2)	0.926 ^a^
no	20 (60.6)	21 (61.8)	
Taking antiepileptics			
yes	2 (6.1)	4 (11.8)	0.414 ^a^
no	31 (93.9)	30 (88.2)	
Taking anxiolytics			
yes	19 (57.6)	21 (61.8)	0.727 ^a^
no	14 (42.4)	13 (38.2)	
Taking antiparkinsonics			
yes	11 (33.3)	9 (26.5)	0.539 ^a^
no	22 (66.7)	25 (73.5)	
Taking antihypertensive drug therapy			
yes	11 (33.3)	8 (23.5)	0.373 ^a^
no	22 (66.7)	26 (76.5)	
Taking hypertriglyceridemia medications			
yes	5 (15.2)	4 (11.8)	0.900 ^a^
no	28 (84.8)	30 (88.2)	
Taking hyperglycemia medications			
yes	0 (0)	5 (14.7)	0.028 ^a,e^
no	33 (100.0)	29 (85.3)	
Smoking status			
current smokers	24 (72.7)	24 (70.6)	0.934 ^a^
ex-smokers/never smokers	9 (27.3)	10 (29.4)	
Alcohol consumption			
yes	2 (6.1)	3 (8.8)	0.972 ^a^
no	31 (93.9)	31 (91.2)	
Physical activity			
inactive	22 (66.7)	24 (70.6)	0.789 ^a^
moderate activity	11 (33.3)	9 (26.5)	
high activity	0 (0) ^d^	1 (2.9) ^d^	
Additional daily energy intake (kJ/kcal)	2259.8/540.1 (2278.2/544.5)	1812.9/433.3 (2279.9/544.9)	0.425 ^c^
The overall SSPI score ^f^	26.5 (13.1)	30.0 (10.9)	0.236 ^b^

IG, intervention group; CG, control group; *n*, number of participants; SD, standard deviation; SSPI scale, Signs and Symptoms of Psychotic Illness scale; ^a^ Pearson Chi-square test (statistically significant *p* < 0.05); ^b^ paired samples t-test (statistically significant *p* < 0.05); ^c^ Mann–Whitney U test (statistically significant *p* < 0.05); ^d^ values not included in the analysis; ^e^ statistically verified that the exclusion of participants taking hyperglycemia medications would not change significantly the blood glucose levels (*p* = 0.143; paired samples t-test); ^f^ the SSPI scale consists of 20 different signs and symptoms of psychotic illness. For each item a score in a range of 0–4 is assigned. The overall score represents the sum of all the subscale scores with 80 being the highest overall score.

**Table 2 nutrients-11-02950-t002:** The impact of the DASH diet vs. the SHD on anthropometric parameters, BP, and biochemical parameters in hospitalized schizophrenic patients with MetS.

Parameter	IG (*n* = 33)			*p* ^b^	CG (*n* = 34)			*p* ^b^	*p* ^c^	*p* ^d^
	Baseline	At 3 Months	EE (%) (95% CI) ^a^		Baseline	At 3 Months	EE (%) (95% CI) ^a^			
	Mean (SD) (95% CI)	Mean (SD) (95% CI)			Mean (SD) (95% CI)	Mean (SD) (95% CI)				
BW (kg)	89.60 (15.81) (83.99–95.21)	87.17 (15.52) (81.67–92.67)	15.7 (14.6–16.7)	<0.001	88.56 (13.97) (83.69–93.43)	86.92 (13.93) (82.06–91.78)	11.8 (10.5–13.1)	0.018	0.776	0.943
BMI (kg/m^2^)	28.95 (5.21) (27.1–30.8)	28.16 (5.04) (26.37–29.95)	15.7 (15.3–16.1)	<0.001	27.47 (3.90) (26.11–28.83)	26.96 (3.95) (25.58–28.34)	12.8 (12.3–13.2)	0.019	0.189	0.281
WC (cm)	109.09 (12.25) (104.75–113.43)	105.55 (12.22) (101.22–109.88)	29.0 (27.7–30.3)	<0.001	106.92 (9.40) (103.64–110.2)	104.47 (9.75) (101.07–107.87)	25.1 (23.7–26.6)	0.002	0.417	0.690
HC (cm)	107.89 (10.55) (104.15–111.63)	106.14 (10.21) (102.52–109.76)	17.2 (16.3–18.0)	<0.001	106.57 (6.90) (104.16–108.98)	105.64 (7.20) (103.13–108.15)	13.0 (11.9–14.1)	0.098	0.546	0.817
WHR	1.01 (0.05) (0.99–1.03)	0.99 (0.06) (0.97–1.01)	27.2 (27.2–27.2)	0.015	1.00 (0.05) (0.98–1.02)	0.99 (0.05) (0.97–1.01)	27.7 (27.7–27.7)	0.006	0.531	0.684
% BF ^e^	28.32 (9.22) (25.05–31.59)	28.34 (8.76) (25.18–31.50)	6.9 (6.1–7.7)	0.165	26.79 (7.97) (23.97–29.61)	27.54 (8.22) (24.62–30.46)	−9.1 (−10.1 to −8.0)	0.185	0.472	0.703
SBP (mmHg)	130.38 (15.53) (124.87–135.89)	126.52 (13.05) (121.89–131.15)	29.6 (24.6–34.7)	0.144	129.41 (19.98) (122.44–136.38)	126.25 (17.90) (120.00–132.5)	17.7 (10.3–25.0)	0.399	0.826	0.945
DBP (mmHg)	85.53 (9.31) (82.23–88.83)	80.23 (9.93) (76.71–83.75)	53.4 (50.1–56.7)	0.004	84.34 (9.70) (80.95–87.72)	82.72 (10.03) (79.22–86.22)	16.1 (12.9–19.4)	0.332	0.610	0.310
TG (mmol/L)	2.15 (0.92) (1.83–2.47)	2.32 (1.21) (1.89–2.75)	−14.1 (−14.4 to −13.8)	0.276	1.81 (0.82) (1.52–2.10)	1.83 (0.79) (1.56–2.10)	−3.0 (−3.3 to −2.7)	0.890	0.118	0.056
HDL-C (mmol/L)	0.92 (0.31) (0.81–1.03)	0.95 (0.30) (0.84–1.06)	−11.1 (−11.2 to −11.1)	0.169	0.93 (0.19) (0.86–1.00)	1.00 (0.29) (0.90–1.10)	−25.0 (−25.1 to −24.9)	0.085	0.859	0.481
LDL-C (mmol/L) ^f^	3.19 (1.11) (2.79–3.59)	3.04 (0.86) (2.73–3.35)	17.0 (16.7–17.3)	0.336	3.07 (0.90) (2.76–3.38)	3.27 (0.78) (3.00–3.54)	−25.5 (−25.8 to −25.2)	0.188	0.646	0.255
TC (mmol/L)	5.08 (1.25) (4.64–5.52)	5.02 (1.07) (4.64–5.40)	5.4 (5.0–5.7)	0.746	4.83 (1.08) (4.45–5.21)	5.09 (1.03) (4.73–5.45)	−25.2 (−25.5 to −24.8)	0.195	0.386	0.795
GLC (mmol/L)	5.59 (0.78) (5.32–5.86)	5.34 (0.53) (5.15–5.53)	47.3 (47.1–47.6)	0.101	6.21 (1.73) (5.61–6.81)	5.69 (1.69) (5.10–6.28)	30.8 (30.1–31.4)	0.146	0.061	0.254

DASH, Dietary Approaches to Stop Hypertension; SHD, standard hospital diet; BP, blood pressure; MetS, metabolic syndrome; IG, intervention group; CG, control group; *n*, number of participants; SD, standard deviation; CI, confidence interval; EE, estimated effect; BW, body weight; BMI, body mass index; WC, waist circumference; HC, hip circumference; WHR, waist-to-hip ratio; % BF, % body fat; SBP, systolic blood pressure; DBP, diastolic blood pressure; TG, triglycerides; HDL-C, high-density lipoprotein cholesterol; LDL-C, low-density lipoprotein cholesterol; TC, total cholesterol; GLC, glucose; ^a^ standardized effect size; represents the ratio of the mean difference and the standard deviation, calculated according to the following formula: (mean difference/standard deviation) * 100; ^b^ comparison of differences within groups using a paired samples t-test; ^c^ comparison of differences between groups at the baseline using an independent samples t-test; ^d^ comparison of differences between groups at three months using an independent samples t-test; ^e^ due to tremor it was not possible to measure % BF at three months for one participant in the IG and both at the baseline and at three months for one participant in the CG; ^f^ both at the baseline and at three months, the LDL-C for one participant in the IG could not be calculated using Fiedewald formula due to TG levels being >4.6 mmol/L.

**Table 3 nutrients-11-02950-t003:** The impact of the DASH diet vs. the SHD on the prevalence of MetS and its features in hospitalized schizophrenic patients with MetS (using Joint Interim Statement diagnostic criteria).

Parameter (95% CI for 1 Proportion) ^a^	IG (*n* = 33)		*p* ^b^	CG (*n* = 34)		*p* ^b^	*p* ^c^	*p* ^d^	OR (95% CI)
	Baseline *n* (%)	At 3 Months *n* (%)		Baseline *n* (%)	At 3 Months *n* (%)				
elevated WC (≥94 cm men; ≥80 cm women)	31 (93.9) (0.80–0.99)	29 (87.9) (0.72–0.97)	0.391	34 (100.0) (0.90–1.00)	32 (94.1) (0.80–0.99)	0.151 ^e^	0.238 ^e^	0.371	1.0 (0.50–2.03)
elevated TG (≥1.7 mmol/L) or pharmacotherapy for elevated TG	27 (81.8) (0.65–0.93)	24 (72.7) (0.54–0.87)	0.378	19 (55.9) (0.38–0.73)	19 (55.9) (0.38–0.73)	1.000	0.022	0.150	1.1 (0.49–2.61)
low HDL-C (<1.0 mmol/L men; <1.3 mmol/L women) or pharmacotherapy for low HDL-C	26 (78.8) (0.61–0.91)	21 (63.6) (0.45–0.80)	0.174	24 (70.6) (0.53–0.85)	17 (50.0) (0.32–0.68)	0.082	0.440	0.260	0.9 (0.38–2.04)
elevated BP (SBP ≥130 mmHg and/or DBP ≥85 mmHg) or pharmacotherapy for elevated BP	8 (24.2) (0.11–0.42)	10 (30.3) (0.16–0.49)	0.580	8 (23.5) (0.11–0.41)	17 (50.0) (0.32–0.68)	0.023	0.945	0.100	1.7 (0.49–5.95)
elevated fasting GLC (≥5.6 mmol/L) or pharmacotherapy for elevated fasting GLC	15 (45.5) (0.28–0.64)	10 (30.3) (0.16–0.49)	0.204	19 (55.9) (0.38–0.73)	18 (52.9) (0.35–0.70)	0.807	0.393	0.060	1.4 (0.51–3.97)
MetS	33 (100.0) (0.89–1.00)	25 (75.8) (0.58–0.89)	0.002 ^e^	34 (100.0) (0.90–1.00)	23 (67.7) (0.49–0.83)	0.0003 ^e^	1.000	0.461	0.9 (0.43–1.87)

DASH, Dietary Approaches to Stop Hypertension; SHD, standard hospital diet; MetS, metabolic syndrome; IG, intervention group; CG, control group; *n*, number of participants; OR, odds ratio; WC, waist circumference; TG, triglycerides; HDL-C, high-density lipoprotein cholesterol; BP, blood pressure; SBP, systolic blood pressure; DBP, diastolic blood pressure; GLC, glucose; ^a^ calculated using Pearson Chi-square test; ^b^ comparison of differences within groups using Pearson Chi-square test or Fisher’s exact test (^e^); ^c^ comparison of differences between groups at the baseline using Pearson Chi-square test or Fisher’s exact test (^e^).;^d^ comparison of differences between groups after three months using Pearson Chi-square test.

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
