# Peer review of "The Effects of the Dietary Approaches to Stop Hypertension (DASH) Diet on Metabolic Syndrome in Hospitalized Schizophrenic Patients: A Randomized Controlled Trial"

_nutrients, 2019, doi:10.3390/nu11122950_

Round 1

Reviewer 1 Report

The revised article is much improved. The authors’ response is adequate for my comments. Thank you.

Author Response

Thank you very much. We are glad we were able to address your concerns. 

Reviewer 2 Report

The manuscript was improved with better methods and conclusions.

I have the following comments:, that you still may improve.

1. The methods stating the sample size calculation is given in lines 103-111. The primary endpoint has then not been described yet, this should be done under statistical analysis.

You did not define your primary endpoint well; I presumed (and suggested) it was the prevalence of MetS at 3 months, and that you wanted to demonstrate a 10% difference in this endpoint. In your sample size calculation (lines 103-111) you inserted that your are looking for a 10% difference in all 'following parameters ... BP, TG, HDL-C, ... and GLC".

This seems incorrect, hope that you agree.

I have done a sample size calculation using the proportions of patients reaching the primary endpoint of MetS at 3 months, with patients in the intervention group having 10% less prevalent MetS than the patients in the control group, with the controle group having an effect of 90% prevalence and the intervention group 80%. In that case, for a >80% power analysis, you would need 219 patients in each group to detect the difference of 10% with significance. What you have done is the sample size for a reduction of 90% to 57%: 33 patients in each group, or the sample size from 99% (no effect) to 72% in the intervention group: 33 patients in each group. You may choose which one you expected, I would choose the 90% to 60% prevalences. (But your study is underpowered to detect a 10% difference). 

Please adapt these calculations in Methods and place sample size calculation under primary endpoint definition.

2. In lines 224-230 (newly inserted) you insert a post-hoc analysis calculation of statistical power. It seems somewhat superfluous, and the argument of being flawed is not a loss of statistical power when you do a per-protocol analysis, but an introduction of bias: who are these patients that drop out of the study, because they could have influenced your results. I would suggest to remove these lines.  

3. IN Methods, it has not been described how MetS is calculated (suggestion: after line 219). It is your primary endpoint, so either describe it or use a good reference.     

4. In Results, lines 303-308, you begin the sentence with "In addition .." . However, these are your main results, so I would suggest to rephrase this: " The impact of the DASH diet on the prevalence of MetS and its features are shown in Table 3".

5. In discussion, you use the words "moreover' (line 358) and 'furthermore' (line 370), but these are words of argument, not of adding sentences together in time. So please begin line 358 with: "After 3 months" and line 370 with "The IG experienced.. "

Author Response

This manuscript is a resubmission of an earlier submission. The following is a list of the peer review reports and author responses from that submission.

Round 1

Reviewer 1 Report

This study is a 3-month parallel-group randomized controlled trial to clarify the relationship between the Dietary Approaches to Stop Hypertension (DASH) diet and metabolic syndrome (MetS). The results showed that there was no difference of the MetS prevalence between the intervention group and the control group at 3 months, even though both groups significantly decreased the MetS prevalence. The authors concluded that nutritional intervention could be a valuable tool for the reduction of MetS prevalence.

This study concept is unique and interesting. However, this study has some limitations. We could not see why this study targeted on schizophrenic patients. I made some comments.

It looks less power study. I cannot see the process how to define this sample size. The authors just wrote the process of a module power analysis. What is the clinical study based on this study? Please add the reference based on the power analysis. Why did the control group decrease the MetS prevalence? This is the main issue why there was no difference of the MetS prevalence between the intervention group and the control group at 3 months. Why did the authors choose 3-month intervention period, not 6 months? I cannot understand the reason why the drop-out participants were hospital discharge. What is the cause of hospitalization? Please write the background cause of hospitalization of the study participants clearly. I suspect that some of the hospitalized schizophrenic patients are difficult to understand this study protocol because the symptoms of schizophrenia are various. The authors wrote ‘’If a participant was deprived of legal capacity, written informed consent was provided from both the participant and his legal guardian.’’, but how many written informed consent were patients themselves or guardian? Why did the study choose schizophrenic patients with MetS? Did the study account for other medications for schizophrenia?

Reviewer 2 Report

The manuscript entitled "The Effects of the Dietary Approaches to Stop Hypertension (DASH) Diet on Metabolic Syndrome in Hospitalized Schizophrenic Patients: A Randomized Controlled Trial " describes the randomization to two diets (DASH versus standard hospital diet) of hospitalized schizophrenia patients with the metabolic syndrome, for the duration of three months. The effect on parameters of the metabolic syndrome is not improved by the DASH diet compared to the hospital diet: in both groups the prevalence of metabolic syndrome decreases, even - non significantly - somewhat more in patients following the hospital diet (DASH: decreased prevalence to 25/33 patients (76%) and in hospital diet: to 23/34 patients (68%)).  

I have the following comments:

the study has the strength that the patients were all hospitalized during the study duration, as was mentioned in discussion. Some patients were excluded before randomization because they were discharged sooner. There were 7 + 4 patients who were discharged after randomization before the end of the study (figure 1), and they were excluded from analysis. This makes the analysis not an intention to treat analysis, but a per-protocol analysis. This should have been foreseen (in Methods), but should also be mentioned in the abstract and in discussion after the main results. in statistical analysis (Methods), the description of the primary endpoint (the endpoint which was used to calculate the sample size with, line 94) is missing. I think that the primary endpoint was the prevalence of Metabolic syndrome, and that the expectation was an extra reduction in the prevalence of metabolic syndrome of 10% by the DASH diet. Please describe this endpoint in Methods under statistical analysis. If you have secondary endpoints, please describe them too.   In the abstract, line 24-25, please provide the final prevalences of the Metabolic syndrome in both groups (76% versus 68%), and not only the P-value. In the abstract, mention as conclusion: line 24-25, "No significant differences in the prevalence of MetS and its features were found between the groups". The following sentence should be removed from the abstract: "In conclusion, this type of nutritional intervention could be a valuable tool for the reduction of MetS prevalence and the amelioration of the overall diet quality in hospitalized schizophrenic patients with MetS". (see comment 5) In both groups, a probably standard 'nutritional counseling program' consisting of 4 lectures was given. The results are interpreted falsely, that this program would have led to the observed decreases in prevalences of MetS. All patients watched TV, but it doesn't mean that the TV decreased the prevalence of MetS. So remove sentences 25-28, lines 292-294 (first lines of discussion),  lines 314-317. At the end of discussion, lines 376-380, a future research perspective is given. Please place these lines before the conclusion, after limitations. In introduction (lines 58-61) ) nor in the discussion, the exact effects of the DASH diet on MetS prevalences in previous studies are given (references 22 and 23 and ? 24). This would certainly help to understand the expected effects and also help in discussing the differences in diets of the control groups (is a DASH diet effective in comparison with a much worse control diet?).   In the table 2, the fourth column contains a 'Estimated effect', which I did not understand, it does not seem to be the difference or the % difference.